# Mycologists and Virologists Align: Proposing *Botrytis cinerea* for Global Mycovirus Studies

**DOI:** 10.3390/v16091483

**Published:** 2024-09-18

**Authors:** Mahmoud E. Khalifa, María A. Ayllón, Lorena Rodriguez Coy, Kim M. Plummer, Anthony R. Gendall, Kar Mun Chooi, Jan A.L. van Kan, Robin M. MacDiarmid

**Affiliations:** 1Botany and Microbiology Department, Faculty of Science, Damietta University, Damietta 34517, Egypt; mkha201@aucklanduni.ac.nz; 2Centro de Biotecnología y Genómica de Plantas, Universidad Politécnica de Madrid (UPM)/Instituto Nacional de Investigación y Tecnología Agraria y Alimentaria (INIA/CSIC), Pozuelo de Alarcón, 28223 Madrid, Spain; mariaangeles.ayllon@upm.es; 3Departamento de Biotecnología Biología Vegetal, Escuela Técnica Superior de Ingeniería Agronómica, Alimentaria y de Biosistemas, Universidad Politécnica de Madrid (UPM), 28040 Madrid, Spain; 4La Trobe Institute for Sustainable Agriculture and Food (LISAF), Department of Animal, Plant and Soil Sciences, School of Agriculture, Biomedicine and Environment, La Trobe University, Bundoora, VIC 3086, Australia; 20852188@students.latrobe.edu.au (L.R.C.); kimplummer1960@gmail.com (K.M.P.); t.gendall@latrobe.edu.au (A.R.G.); 5Australian Research Council Research Hub for Sustainable Crop Protection, La Trobe University, Bundoora, VIC 3086, Australia; 6The New Zealand Institute for Plant and Food Research Limited, Auckland 1025, New Zealand; karmun.chooi@plantandfood.co.nz; 7Laboratory of Phytopathology, Wageningen University, 6708 PB Wageningen, The Netherlands; jan.vankan@wur.nl; 8School of Biological Sciences, The University of Auckland, Auckland 1010, New Zealand

**Keywords:** model system, *Botrytis cinerea*, mycovirus

## Abstract

Mycoviruses are highly genetically diverse and can significantly change their fungal host’s phenotype, yet they are generally under-described in genotypic and biological studies. We propose *Botrytis cinerea* as a model mycovirus system in which to develop a deeper understanding of mycovirus epidemiology including diversity, impact, and the associated cellular biology of the host and virus interaction. Over 100 mycoviruses have been described in this fungal host. *B. cinerea* is an ideal model fungus for mycovirology as it has highly tractable characteristics—it is easy to culture, has a worldwide distribution, infects a wide range of host plants, can be transformed and gene-edited, and has an existing depth of biological resources including annotated genomes, transcriptomes, and isolates with gene knockouts. Focusing on a model system for mycoviruses will enable the research community to address deep research questions that cannot be answered in a non-systematic manner. Since *B. cinerea* is a major plant pathogen, new insights may have immediate utility as well as creating new knowledge that complements and extends the knowledge of mycovirus interactions in other fungi, alone or with their respective plant hosts. In this review, we set out some of the critical steps required to develop *B. cinerea* as a model mycovirus system and how this may be used in the future.

## 1. Introduction

Mycoviruses are master puppeteers of their fungal hosts, but how they pull these strings is largely unclear. Mycoviruses can convert a fungus from a pathogen to a beneficial endophyte [1], act as a biocontrol of pathogenic fungi [2,3], and conversely may limit the efficacy of novel antifungal agents [4]. Yet, mycoviruses are generally under-described in genotypic and biological studies. To achieve a greater understanding of the diversity of mycoviruses, their interactions and impacts on their host (whether alone or in multiple infections within a single isolate), we propose that a single model fungus host species is researched in great depth. *Botrytis cinerea* is an ideal model mycovirus system in which the international community can develop a deep understanding of mycovirus epidemiology including diversity, impact, and associated cellular biology of the host and virus interactions.

## 2. Mycoviruses

Mycoviruses are extra-chromosomal strands of nucleic acid that are replicated by their fungal host. There is a wide world of sequence space that has resulted in genomes that are either RNA or DNA molecules (single or multisegmented) that may be packaged or naked. Modern sequencing and analysis technologies have enabled the direct identification of thousands of new virus species, including new mycovirus species, genera, and even families [5]. Most mycoviruses have linear RNA genomes whether they are double-stranded (10 classified families and other unclassified mycoviruses), single-stranded positive sense (12 classified families that include a satellite RNA and other unclassified mycoviruses) or single-stranded negative sense (three classified families and other unclassified mycoviruses). A single family of mycoviruses have circular DNA genomes. Excellent reviews describe these mycoviruses, their taxonomic diversity, and the impact on the fungi they infect [6,7,8,9,10,11]. It is this diversity and their impact on their fungal host that is imperative to understand as we are now at a time of prevalent fungicide resistance [12,13] and climate-driven fungal outbreaks [14,15]. We urgently need knowledge-based, new-era control strategies for pathogenic fungi. 

A model fungus that is amenable to research and available globally, can be used to fill the knowledge gap by addressing urgent and fundamental biological questions. For instance, What is the diversity of mycoviruses infecting a single fungal species, internationally? How prevalent are mycoviruses in fungi growing in the environment? Are particular mycoviruses antagonistic or incompatible with viruses of other species? What antiviral defense mechanisms are operative in fungi, how do viruses counteract them, and what impact does this have on exogenous application of double-stranded RNA (dsRNA)? How do mycoviruses move between vegetatively incompatible fungal isolates or species and into plants? Do mycoviruses exploit movement protein activity? What changes (epigenetic, biochemical, physiological, etc.) in mycovirus-infected fungi affect fungal pathogenicity? Why do fungi continue to support so many mycoviruses? Do some mycoviruses specialize to a small number of hosts, and conversely do some mycoviruses have a broad host range? How transferable are learnings from one fungus–mycovirus complex to another? Will mycoviruses provide new risks or benefits under a changing climate?

## 3. *Botrytis cinerea*

*Botrytis cinerea* (previously also known as *Botryotinia fuckeliana*) is an air- and insect-borne plant pathogen that attacks more than 1400 mainly dicotyledonous plant hosts worldwide, including many important crop species [16,17]. It is one of the most cosmopolitan and damaging phytopathogenic fungi and many isolates have developed resistance to one or more fungicides [18]. Few control options exist for this pathogen. It has already been demonstrated to be a suitable experimental system to study many developmental and physiological processes in fungi [19], including circadian rhythms [20], light perception and response [21], vegetative incompatibility [22], sexual reproduction [23] and the induction of host cell death [24]. Genomes of multiple isolates have been sequenced [25,26,27,28], and the fungus is genetically malleable by making knockout or complementation mutants, as well as carrying out allele-specific gene editing [29,30,31]. There is a large scientific community with isolate collections around the world [32,33]. Finally, *B. cinerea* has one of the most numerous and divergent mycoviromes reported in scientific literature. 

### 3.1. Botrytis cinerea Hosts a High Number and Diversity of Mycoviruses

Mycoviruses and also plant viruses have been identified present within *B. cinerea* (Table 1 and Table 2). To date, collections of isolates from various regions of the world (Southern America (Chile), Asia (China and Lebanon), Europe (Germany, Italy and Spain) and Oceania (Australia and New Zealand) have already been searched for mycoviruses of *B. cinerea* and these studies have unveiled a quite complex mycovirome. The sequence description of full-length genomes is often published alongside other data (either sequence data of other viruses or some biological description). Sometimes partial genomes are published either purposefully or unknowingly, especially in the case of multi-segment viruses in which the number of total genome segments is unknown. The virus sequence is often published with a proposed name for the virus and a proposed virus family, and/or genus (existing or new) into which it fits depending on the specifications outlined by the International Committee for the Taxonomy of Viruses [34]. While pairwise sequence similarity and phylogenetic relationships are predominant criteria for classification, genome composition and expression pattern, capsid structure, presence or absence of a lipid envelope, host range and pathogenicity are also considered as required. Of note, each virus family, genus and species has their own criteria depending on the sequence variation found among members. 

A summary of the approved viruses is typically published in the Archives of Virology and on the ICTV website [35]. Some long-known viruses have remained unclassified and are considered distinct from existing classified viruses due to their unique sequence and/or biological features [36]. Other unclassified viruses are those for which sequence data but not information on biology or pathogenicity are available and no proposal has passed through the ICTV classification process, although sequence data alone are sufficient for classification [37]. An overview of each classified family is reviewed by Ayllón and Vainio [7] and Hough et al. [38]. Discovered in *B. cinerea*, GenBank contains no less than 113 viruses distributed among 21 viral families that represent the different types of genomes (Appendix A, Table 1 and Table 2). Among sequenced *Botrytis* mycoviruses, most positive-sense single-strand RNA (ssRNA) viruses were classified into several families (*Alphaflexiviridae*, *Gammaflexiviridae*, *Hypoviridae*, *Narnaviridae*, *Mitoviridae*, *Fusariviridae*, *Botourmiaviridae*, and *Endornaviridae*) [39,40,41,42,43]. Most dsRNA viruses were assigned into families *Partitiviridae*, *Totiviridae* and *Quadriviridae*, and the genus *Botybirnavirus* [42,44,45,46]. The negative-sense ssRNA viruses are classified into the family *Mymonaviridae* [47]. The single-stranded (ssDNA) DNA viruses are classified into the family *Genomoviridae* [43,48,49]. In addition, several sequenced mycoviruses remained unclassified [42,43,50].
viruses-16-01483-t001_Table 1Table 1Summary of virus families represented by viruses detected in *Botrytis cinerea*.GenomeFamilyNo. of SpeciesDistributionTypeSegmented?dsRNA√*Partitiviridae*4Chile, China, Italy, Spain√*Botybirnaviridae **(proposed)4Chile, Pakistan, Spain√*Quadriviridae*2China, Spain
*Totiviridae* *5China, Italy, Pakistan, Spain, USA
Unclassified **3China, Colombia, Italy, Spain(+)ssRNA
*Togaviridae-*related1Italy, Australia
*Botourmiaviridae*20China, Italy, Pakistan, Spain, Australia
*Deltaflexiviridae*4China, Pakistan, Spain, Australia
*Endornaviridae*4China, Pakistan, Italy, Australia
*Fusariviridae*9China, Italy, Pakistan, Spain, Australia
*Hypoviridae*6China, Italy, Pakistan, Spain, Australia
*Hypoviridae* satellite ***1China, Russia, Spain
*Mitoviridae* ****14China, Italy, Pakistan, Russia, Spain, Australia
*Tymoviridae*-related1China, Pakistan, Spain
*Narnaviridae*1Spain√*Splipalmiviridae* (proposed)5China, Pakistan, Spain
*Mycotombusviridae* or*Ambiguiviridae*(proposed)4China, Pakistan, Spain
*Gammaflexiviridae*1France, New Zealand, Spain, Australia
*Alphaflexiviridae*1New Zealand
Unclassified1Italy(−)ssRNA√*Phenuiviridae*1Spain
*Mymonaviridae*9China, Italy, Pakistan, Spain, Australia
*Peribunyaviridae-*related2Italy, Pakistan, Spain
Unclassified*8*China, Italy, Pakistan, Spain*ssDNA**√**Genomoviridae*1China, Italy, New Zealand, Spain*113*

113
* Also detected associated with *Vitis vinifera* cv Syrah. ** Also detected associated with *Solanum lycopersicum*. *** Also detected associated with *Vitis vinifera* cv Rkatsiteli. **** Also detected associated with *Vitis vinifera* cv Rkatsiteli and *Erysiphe necator*.
viruses-16-01483-t002_Table 2Table 2Viruses of other fungi detected in *Botrytis cinerea*.GenomeGenusVirusLocationReference
dsRNA
*
Botybirnavirus
*
Botrytis porri botybirnavirus 1 (BpBV1)
Spain[42]*Unclassified*Sclerotinia sclerotiorum dsRNA mycovirus L (SsNsV-L)Spain and Australia[40,42]
(+)ssRNA
*
Deltaflexivirus
*
Sclerotinia sclerotiorum deltaflexivirus 2 (SsDFV2)
Spain, Italy and Australia[42]*
Umbravirus
*
Sclerotinia sclerotiorum umbra-like virus 2 (SsUV2)
Spain and Italy[42]*
Umbravirus
*
Sclerotinia sclerotiorum umbra-like virus 3 (SsUV3)
Spain, Italy and Australia[42]*
Betahypovirus
*
Sclerotinia sclerotiorum hypovirus 1 A (SsHV1A)
Spain and Italy[42]*
Betascleroulivirus
*
Pyricularia oryzae ourmia-like virus 2 (PoOLV2)
Italy[42]*
Duamitovirus
*
Sclerotinia sclerotiorum mitovirus 3 (SsMV3)
Spain and Italy[40,42]*
Unuamitovirus
*
Sclerotinia sclerotiorum mitovirus 4 (SsMV4)
Spain and Italy[42]

### 3.2. Incidence of Mycoviruses in Botrytis cinerea

Mycovirus prevalence in *B. cinerea* has been reported to range from 0.8% to 100% across various countries (Table 3). This incidence can fluctuate depending on the sampling location, methodologies employed (for instance determining the incidence of specific mycovirus(es), or determining the whole mycovirome by RNA sequencing), and host/virus combination [51,52,53]. Fungal isolates stored for a long period had a lower prevalence of mycoviruses compared to the fresh-cultured field isolates [42,53]. 

To date, many studies have focused on *B. cinerea* mycovirome determination using RNA-Seq, and two studies have shown that isolates from Italy, Spain or Australia are typically infected with one or more mycoviruses with 83–100% infection incidence [42,54] (Table 3). The most common detection technologies are reverse transcription polymerase chain reaction (RT-PCR) and traditional dsRNA band profiling, which have been used to estimate the presence of specific mycoviruses, or of mycoviruses in general. Howitt et al. [55] used the dsRNA profiling method and reported an incidence of 72% of mycoviruses in New Zealand *B. cinerea* isolates. In contrast, in Chile, the incidence was 3% [56]. Some studies have reported a lower incidence of a specific mycovirus in isolates of *B. cinerea*; however, that does not exclude the possibility that the fungal isolates are infected with other distinct mycoviruses.
viruses-16-01483-t003_Table 3Table 3Mycovirus incidence in *Botrytis cinerea* isolates reported in published studies across the world using different discovery methods.IncidenceNo. of Isolates/SamplesDetection MethodLocation, Fungus Host [If Reported], Field/Cultured IsolateReference100%29 pools (total 248 isolates)RNA-SeqItaly and Spain, *Vitis vinifera*, field[42]93%29Botrytis cinerea mitovirus 1 specificRT-PCR and Sanger sequencing *Spain, *Capsicum annuum*, *Cucumis sativus*, *Cucurbita pepo*, *Solanum lycopersicum*, *Solanum melongena*,*Phaseolus vulgaris*, *Vitis vinifera*, field[53]83%24RNASeqAustralia, a wide range of plants, cultured[54]72%200dsRNA purification *New Zealand, *Cucumis sativus*, *V. vinifera*, *Solanum lycopersicum*, *Fragaria* × *ananassa*, *Phaseolus vulgaris*, *Rubus fruticosus*, cultured[55]55%96dsRNA purification *Spain, *Capsicum annuum, Cucumis sativus*, *Cucurbita pepo*, *Solanum lycopersicum*, *Solanum melongena*,*Phaseolus vulgaris*, *Vitis vinifera*, field[53]29%87Botrytis virus X RT-PCRNew Zealand, a wide range of plants, cultured[46]27.8%248Botrytis cinerea ssDNA virus 1RT-PCRSpain and Italy, *Vitis vinifera*, field[43]16%87Botrytis virus F RT-PCRInternational, a wide range of plants, cultured[46]14%84Botrytis virus F RT-PCRInternational, a wide range of plants, cultured[57]4.8%21dsRNA purification *China, wide range of plants (suggestion)[58]3%30dsRNA purification *Chile, *Malus domestica*, *Pyrus*, *Rubus idaeus*, *Vitis vinífera*, field[56]2%500*Genomoviridae* rolling-circle amplification and high throughput sequencing of productNew Zealand, a wide range of asymptomatic plants, cultured[49]0.8508Botrytis cinerea mymonavirus 1RT-PCRChina[59]* dsRNAs purified using CF11 cellulose.

## 4. The Dual Challenges of Mycovirology: Virus Description and Biology

Fungal virus research is one of the fastest growing fields, as researchers around the world are interested in exploring new types of viruses and their potential effects on the host. Astonishing mycovirus-mycovirus interactions and mycovirus–fungus interactions with the wider ecosystem are being discovered. One driver for studying fungal viruses is the attempts to use them in biological management of fungi in the case where the mycovirus can weaken the pathogenic fungal host. This opportunity is timely as it is coincidental with the global trend to reduce the use of fungicides to combat fungi due to environmental and consumer reasons. Hypovirulence in general (defined as the reduced ability of the fungal host to cause disease) is associated with some mycoviruses that are classified with a range of families including *Hypoviridae*, *Mitoviridae*, *Narnaviridae*, *Fusariviridae*, *Endornaviridae*, *Mymonaviridae*, *Partitiviridae*, *Totiviridae*, *Megabirnavirus*, *Botourmiaviridae*, *Spinareoviridae*, *Rhabdoviridae*, *Alphaflexiviridae*, *Deltaflexiviridae*, *Solemoviridae*-related, proposed family *Ambiguiviridae* or *Mycotombusviridae* and *Genomoviridae* [6,60,61].

### 4.1. Mycovirus Description

Harnessing bioinformatics tools as well as next-generation sequencing has helped facilitate the study of fungal viruses and the discovery of many different types in a short period. A range of bioinformatics pipelines have been used to identify and distinguish mycovirus sequences from their hosts or other sequences [62]. Like much virus discovery, identification is based on homology to known mycoviruses; therefore, larger databases enable further discoveries of mycoviruses with wider homology disparity to those currently described. Essential to all virus association with host is deconvolution of the holobiome. For instance, many mycoviruses have been described in association with insects despite their hosts being fungi [63]. By contrast, mycoviruses have also been described to be hosted by plants which may serve as bridges or aids between host fungal isolates or even species [64]. Care must be taken in the attribution between ‘association’ and bona fide ‘host’ and therefore whether novel virus sequences are mycoviruses or not. 

### 4.2. Mycovirus Biology

Interestingly, Botrytis virus F and Botrytis virus X have no significant effects on the virulence of *B. cinerea*, although the BVF infectious clone reduced the growth of a particular *B. cinerea* strain isolated from pepper [65]. Several other mycoviruses have been associated with hypovirulence in *B. cinerea*, indicating that it is feasible to use mycoviruses in biocontrol of the fungus. For instance, Botrytis cinerea mitovirus 1 [60], Botrytis cinerea endornavirus 1 [46], Botrytis cinerea hypovirus 1 and Botrytis cinerea fusarivirus 1 [66], Botrytis cinerea mymonavirus 1 [61], Botrytis cinerea partitivirus 2 [67], and Botrytis gemydayirivirus [49] were all reported to be associated with hypovirulence of *B. cinerea*. 

One of the possible barriers to establish a biological control strategy for *B. cinerea* based on the use of mycoviruses is the large number of vegetative compatibility groups (VCGs). The limited occurrence of isolates displaying the same VCG suggests that sexual recombination occurs in field populations of *B. cinerea*. However, in Spain, for instance, apothecia (sexual fruiting bodies) of *B. cinerea* are infrequently found in nature and the large proportion of field isolates infected by mycoviruses [39,40,42,50,53] suggests that sexual reproduction is infrequent, indicating that vegetative incompatibility probably is not an impediment for mycoviral exchange. Moreover, mycoviruses infecting *B. cinerea* have been identified that appear capable of overcoming vegetative incompatibility in horizontal transmission [67]. In nature, transmission of mycovirus between *B. cinerea* from different VCGs may occur predominantly in planta, as demonstrated recently for mycovirus transmission between distinct VCGs of *Sclerotinia sclerotiorum* [68]. Remarkably, Cryphonectria hypovirus 1 infection enzymatically alters the volatile profile of its fungal host to make both the host (and virus) more attractive to insect vectors [69].

Several systems have been used as a tool for deciphering and understanding molecular mechanisms involved in fungal-host interaction and fungal pathogenesis, with some examples provided here. A study in *Fusarium oxysporum* provided microscopic evidence about the effect of a hypovirulence-inducing mycovirus on the pattern of plant colonization by its fungal host [70]. Moreover, changes in the phenotype or virulence of the mycovirus-infected fungi have been correlated with a reprogramming of the fungal transcriptome. For instance, Cryphonectria hypovirus 1 infection down-regulates the expression of the heat shock protein Hsp24 in *Cryphonectria parasitica* and is associated with the fungal stress response and virulence [71], but increases the expression of Dicer-like 2 and Argonaut-like 2, required for RNA silencing in response to viral infection [72]. In the interaction of *Fusarium graminearum*-FgV-ch9 (Fusarium graminearum virus China 9), a novel molecular determinant for symptom development in the virus-infected fungus was proposed, linking hypovirulence to the presence of a viral structural protein and a putative poly(A)-binding protein [73]. RNA-Seq–based, genome-wide expression analysis revealed distinct patterns of *F. graminearum* transcriptomes in response to individual infections by four dsRNA mycoviruses. The fungal host transcriptome was more often affected by Fusarium graminearum virus 1 (FgV1) and FgV4 infections than by FgV2 and FgV3 infections, resulting overall in down-regulation of host genes required for cellular transport, RNA processing and ribosome assembly [74]. The infection of *S. sclerotiorum* with Sclerotinia sclerotiorum debilitation-associated RNA virus (SsDARV) induced the differential expression of genes representing a broad spectrum of biological functions including carbon and energy metabolism, protein synthesis and transport, signal transduction and stress response [75]. Additionally, metabolic processes, biosynthesis of antibiotics, and secondary metabolites were the most affected categories upon SsHADV-1 infection, and one third of the differential expressed genes were involved in the signal transduction mediated by Ras-small G protein [76]. More recently, Wang et al. [77] showed that Bipolaris maydis partitivirus 36 infection in *Bipolaris maydis* significantly up-regulated membrane-related genes, but significantly down-regulated genes related to membrane transport, synthesis of toxins, cell-wall-degrading enzymes, carbohydrate macromolecule polysaccharide metabolic, and catabolic processes. Similar results were found in a hypovirulent strain of *Rhizoctonia solani* infected with Rhizoctonia solani partitivirus BS-5. The transcriptomic analysis revealed that the number of differentially expressed genes involved in cell-wall-degrading enzyme genes was reduced in the infected strain and associated with the hypovirulence mediated by the partitivirus [78]. These studies have improved the knowledge of the mechanisms involved in the interaction of mycoviruses with their fungal hosts, but at the same time, and perhaps more importantly, provide new clues to develop more specific control strategies of the fungal pathogens. Currently, no such studies have been reported in *B. cinerea* in the interaction with mycoviruses, but efforts to initiate such studies are ongoing in laboratories of the authors.

### 4.3. Antiviral Suppressors Encoded by Mycoviruses

#### 4.3.1. RNAi Activity

RNA interference (RNAi, also known as RNA silencing) is an antiviral activity common to multi-cellular organisms. The antiviral defense is predominantly executed by two core enzymes, namely Dicer and Argonaute (AGO) that cleave dsRNA or sequence-specific target RNA, respectively. Viruses produce dsRNA either as a replication intermediate (in positive- or negative-sense ssRNA or dsRNA genome viruses) and/or through production of dsRNA structures formed from base-pairing within a single RNA molecule such as a genomic RNA, or for DNA viruses, a genome copy, or sub-genomic RNAs. The host-encoded Dicer (or in plants Dicer-like, DCL) protein recognizes and cleaves dsRNA into smaller units (small RNAs, sRNA, 19–24 nt long). One sRNA strand (the guide strand) is incorporated into AGO and guides the enzyme to a target RNA through sequence complementarity. These sRNAs bound to AGO (or AGO-like in fungi) are incorporated into the RNA-induced silencing complex (RISC) and AGO can either cleave the target RNA at the midpoint of the recognized sequence or block translation of the mRNA target. Both AGO outcomes cause reduced expression of the protein encoded by the target mRNA. Cleaved target RNA fragments can serve as primers for RNA-dependent RNA polymerase (RdRp, Rdp, or QDE1 in some fungi), generating more dsRNA and amplifying the RNAi cycle [79,80]. Additionally, sRNAs can direct the methylation of matching DNA sequences. Depending on the species, other proteins, (e.g., DRMs, SDS3 and RDRs) are also involved in aiding the core enzymes, amplifying the dsRNA signal or moving small interfering RNAs (siRNAs) throughout the organism. 

The RNAi pathway also forms a critical negative gene regulation pathway. Host-encoded dsRNA, termed pri-microRNA precursors, generate microRNAs (miRNAs) through action of Dicer that target and inhibit host-encoded target mRNAs through base complementarity in the context of AGO. In this manner, many genes of higher-order organisms are down-regulated at critical periods of development to create specialized tissues and organs. RNAi also plays crucial roles in various biological processes in fungi, including the control of transposon movement, ascus and meiocyte formation, as well as meiotic silencing [81,82,83,84].

Interestingly, RNAi can be activated by dsRNA from both endogenous and exogenous sources. This has been exploited in the development of spray-induced gene silencing (SIGS), and host-induced gene silencing (HIGS), respectively, as approaches to inhibit fungal pathogenesis [85,86,87,88,89]. 

#### 4.3.2. Fungal RNAi Activity

Several studies have shown the role of RNAi as an antimycoviral defense in plant pathogenic fungi, processing the mycoviral RNA into small RNAs that can be detected in small RNA-Seq assays. The first two studies to show this fact were performed by Zhang et al. [90] and Hammond et al. [91] in *C. parasitica* and *Aspergillus nidulans*, respectively. The same was shown in other fungi such as *Magnaporthe oryzae* [92], *F. graminearum* [93], *Rosellinia necatrix* [94], *S. sclerotiorum* [95] or *B. cinerea* [40]. Recently, DCL2 but not AGO2 was shown to be required to confer antiviral defense against some but not all mycoviruses of *C. parasitica* [96]. Within *B. cinerea*, there are two DCL genes [31,97,98] and three RDR genes [99] which have been functionally studied, while three AGO genes remain to be functionally characterized. Transfection of wild-type and ∆*Bcdcl2* mutant *B. cinerea* lines with Botrytis virus F (BVF) did not cause detectable alteration in fungal growth or virulence. The ∆*Bcdcl2* mutant *B. cinerea* mutant remained capable of gene silencing and appeared not to be affected in BVF titers [99]. There are conflicting reports regarding the role of DCL genes in *B. cinerea* development and virulence. While Weiberg et al. [98] reported reduced virulence for a ∆*Bcdcl1/*∆*Bcdcl2* double mutant that also displayed morphological aberration, Qin et al. [31] reported that multiple independent (newly generated) ∆*Bcdcl*1*/*∆*Bcdcl*2 double mutants, in the same recipient strain background, neither showed altered morphology nor reduction in virulence on four different plant hosts, as compared with wild-type.

A recent study by Cheng et al. [99] characterized the functions of BcRDR1 and BcRDR2, which are orthologs of the *Neurospora crassa* genes SAD-1 and QDE-1, respectively. Two independent ∆*Bcrdr*1 mutants showed a reduction in virulence that was associated with reduced production of fungal retrotransposon-derived small RNAs that were proposed to participate in silencing of defense responses in the host plant. No reduced virulence was observed in ∆*Bcrdr*2 mutants, and homokaryotic mutants in the *Bcrdr*3 gene were not obtained.

#### 4.3.3. Virus-Encoded Suppressors of RNAi (VSRs) 

VSRs counter the RNAi activity of the host [100]. Some viruses encode a single VSR whereas others may encode several, each with a slightly different activity, e.g., inhibiting Dicer, AGO or binding dsRNA or siRNAs and thereby preventing their participation in the RNAi mechanism [101]. VSRs were initially and extensively identified in plant infecting viruses although they have also been identified in viruses that infect animals [102,103]. Some mycovirus-encoded suppressors of RNAi have also been described (see below). Problematically, plant assays are the mainstream for identifying VSR activity regardless of the source of the virus. However, RNAi functioning is not identical across all multi-cellular organisms. 

#### 4.3.4. Mycovirus-Encoded Suppressors of RNAi (VSRs) 

In general, fungal VSRs lack conserved sequence motifs, as they encode a variety of different proteins across different viral species, families, and fungal hosts. This variability within VSRs makes it challenging to use bioinformatics tools to confirm the presence of VSR activity [104]. Instead, methods commonly employed to identify VSRs in plants, humans, or insects are also applied to fungi. These standard techniques include transient expression assays, silencing reversal assays, and stable expression assays [93]. Alternatively, indications of VSR activity within mycovirus-infected compared with uninfected fungi has been detected using changes in small RNA profiles; fungal genes involved have been discovered using wild-type or isolates carrying mutated RNAi-mechanism genes. VSR activity has been observed in multiple mycoviruses and plant viruses from various families whose members include those that have been identified within *B. cinerea* (Table 4).

VSR activity has been documented in plant viruses of the families *Alphaflexiviridae, Tymoviridae*, *Tombusviridae* (related to *Ourmiavirus* within *Botourmiaviridae*) and *Phenuiviridae* as well as human viruses with the family *Togaviridae* (related to *Mycotymoviridae*) (Table 4). Whether related mycoviruses also encode VSRs or whether these are not required in fungi remains unknown. 

A mutant *dcr2* (*Bcdcl2*) made no difference to host phenotype or virulence; however, the expression of dcl 1 (in Δdcr2 mutant) and dcl2 (in wild-type) were suppressed at early stages of BVF (*Gammaflexiviridae*) infection of *B. cinerea*, demonstrating that BVF encodes a VSR [97]. 

Mycoviruses within several families have demonstrated VSR activity although not yet in *B. cinerea*. In the family *Fusariviridae*, P2 gene of FgV1 was found to play a role as a VSR by suppressing transcriptional up-regulation of the key enzyme genes such as FgDICER2 and FgAGO1 [113]. Indirect evidence indicated that RNA silencing served as an antiviral defense mechanism in *C. parasitica* by observing that the papain-like protease p29 encoded by Cryphonectria hypovirus 1-EP713 (*Hypoviridae*) was able to suppress gene silencing in the fungus and in the plant [114,115]. VSRs able to reduce the accumulation of siRNA were detected in a member of totiviruses (*Totiviridae*). Moreover, the coat proteins (CPs) of Tulasnella partitiviruses 2 and 3 exhibit VSR activity [117]. Some dsRNA mycoviruses have demonstrated VSR activity including member within the family *Partitiviridae* although the mechanism is not known [117]. 

It is not known yet whether endornaviral genomes can express proteins with VSR activity. Endornavirus-derived small interfering RNAs (siRNAs) have been detected in host plants infected with several endornaviruses and several endornaviral genomes were assembled from siRNA [118,119], indicating successful recognition of the viral RNA by the hosts’ RNA silencing machinery. No reports are indicating that viruses within *Deltaflexiviridae*, *Gammaflexiviridae*, *Narnaviridae*, *Mitoviridae*, *Botourmiaviridae, Mymonaviridae*, *Quadriviridae*, *Botybirnaviridae*, *Peribunyaviridae* or *Splipalmviridae* are capable of encoding VSRs.

## 5. *Botrytis cinerea* Serves as a Perfect Experimental System

*B. cinerea* is the most extensively studied necrotrophic plant pathogenic fungus; *B. cinerea* is fast-growing, making host infection and in vitro studies relatively easy [19]. There are numerous collections of *B. cinerea* from several continents, including virus-free and virus-containing strains; *B. cinerea* field isolates are frequently found to host multiple mycovirus species and are able to be ‘cured’ of their mycoviruses by rounds of single spore isolation, regrowth from protoplasts, or hyphal fusion (anastomosis) (unpublished data Spanish and Australian groups, [120,121]). The approaches all facilitate the production of isogenic strains carrying or lacking mycoviruses to determine the impact of mycovirus presence/absence on pathogenicity and other biological processes. *B. cinerea* produces both sexual and asexual spores in vitro, as well as persistent survival structures (sclerotia) that are easily stored at 4 °C or room temperature. Asexual reproduction is the primary mode of survival and reproduction in *B. cinerea* growing under field conditions. This is ideal for the obligate mycoviral parasites that require a fungal host for survival. Sexual fruiting bodies of *B. cinerea* have rarely been observed in the field, even though isolates are able to reproduce sexually in vitro under laboratory conditions and one study provided evidence for regular occurrence of sexual reproduction within *B. cinerea* populations in France [122]. This lack of (frequent) sexual sporulation in the field [123] may in part explain the relatively high abundance of mycoviruses in field isolates, as mycoviruses may be lost during the development of sexual spores (ascospores). It is possible that mycoviruses interfere with the biology of *B. cinerea*, promoting the attraction of insect vectors (as is reported in some other fungal systems), altering the competitive saprophytic ability [124] or inhibiting sexual reproduction in the field [124,125]. However, any hypotheses regarding the efficacy and limitations of mycovirus transmission in *B. cinerea* have yet to be fully investigated. 

*B. cinerea* can serve as an excellent model system for studying mycovirus biology and their influence on the physiology of the fungal host. Genomes of multiple *B. cinerea* isolates have been sequenced [26,27,28], including a finished community-annotated reference genome. The fungus is genetically amenable to modification through knock-out or knock-in mutants in multiple genes (the current maximum is 21 genes knocked out in a single strain), as well as carrying out allele-specific gene editing [29,30,31]. Reporter lines with fluorescent markers are available and transcriptome, metabolome and proteome data are published for multiple isolates [126]. Antibodies raised against overexpressed BVX CP were employed in immunofluorescence microscopy of ultrathin hyphal sections by Boine et al. [121] who identified the intracellular localization of the BVX in *B. cinerea* to be near the cell membranes and walls of septal pores and hyphal tips. 

## 6. Other Fungal Model Systems for Mycoviruses

Mycovirus research has occurred at pace in other fungal hosts; however, each of these hosts has one or more drawbacks for use as an international model for mycovirus research compared with *B. cinerea*. 

### 6.1. Saccharomyces cerevisiae

The single cell yeast *Saccharomyces cerevisiae* is used as model for understanding many biological processes in eukaryotes [127], and to study biological processes of both plant and animal viruses [128]. It has a small genome can live in either haploid or diploid states. About 50% of isolates of *S. cerevisiae* have been reported to carry dsRNAs from the mycovirus families *Totiviridae* and *Partitiviridae*, as well as satellite dsRNAs that are associated with killer toxin production [129]. Although this model has major benefits for studying genetics and interactomes, the major drawbacks of using *S. cerevisiae* as a model for mycoviruses are its lack of representation for multicellular fungi, its loss of RNAi [130], and its lack of infection of plants. 

### 6.2. Cryphonectria parasitica

Excellent progress has been made in understanding the biology of the interaction between mycoviruses, *Cryphonectria parasitica* and the various plant hosts (American and European chestnut trees) [131]. In particular, mycoviruses that are associated with reduced virulence of the phytopathogenic host have been trialed and used as biocontrol agents. *C. parasitica* has many of the features of the desired attributes a model fungus—such as being biologically and genetically tractable including a growing list of biological resources and molecular tools [131]. For instance, wild-type *C. parasitica* and its *dcl*2 mutants have been used as a model system to study viruses that originate from other fungi [132]. Importantly, *C. parasitica* has been used to describe the suppressor of RNA silencing activity of p29 that is encoded by Cryphonectria hypovirus 1 and to determine the fungal host proteins involved in antiviral activity including distinct antiviral contributions under different mycovirus infections [96,115,133]. 

Chestnut blight/hypovirus is limited as an international model system due to the narrow host range of the *C. parasitica* woody host and its limited distribution around the world. Originally from Eastern Asia, in the past it has invaded Europe, North America and Australia; therefore, it is on the World Invasive Species Database [134] and is a regulated or quarantine pest in many countries. It would therefore be associated with strict physical containment constraints, greatly complicating its use as a model organism [135,136,137]. By contrast, the range and variety of the mycovirus species in *B. cinerea*, the broad host range of *B. cinerea*, and the ease of manipulation of *B. cinerea* biology is an advantage over the *C. parasitica*/mycovirus/chestnut system. 

### 6.3. Fusarium graminearum 

*F. graminearum* is a widely dispersed fungus that infects staple cereals such as wheat, barley, rice, oats and corn, resulting in head or ear blight along with the production of a mycotoxin that is harmful to humans and livestock. Whole genome sequencing has been completed for this species complex that includes at least 16 monophyletic species [138]. A range of genetic tools and collections exist for *F. graminearum* [19]. Mycoviruses of at least 12 mycovirus families have been detected within members of the *Fusarium* genus although only five families (*Alternaviridae*, *Chrysoviridae*, *Fusagraviridae*, *Hypoviridae* and *Genomoviridae*), and an unclassified mycovirus have representative mycoviruses detected in *F. graminearum* [139,140]. This includes three mycoviruses classified within *Chrysoviridae* that are promising biocontrol agents associated with hypovirulence of their host [141]. By contrast, Fusarium graminearum hypovirus 1 (FgHV1) is not associated with reduced pathogenicity of its *Fusarium* host. 

Important studies of mycovirus–host interactions have described both transcriptional and protein reprogramming of the infected host [141]. Fusarium graminearum gemytripvirus 1 has been shown to be associated with severely reduced colony growth and hypovirulence [139], and recently, a virus-induced gene silencing (VIGS) vector based in this mycovirus have been developed and shown to be a promising biocontrol agent to protect wheat from *F. graminearum* and mycotoxin contamination [142]. Interesting new findings in *F. graminearum* have shown that mycovirus-infected strains are more attractive to some fungivores as well as having a reduction in the levels of the mycotoxin, deoxynivalenol [120]. This has implications for many necrotrophic fungi that produce mycotoxins, but more interestingly can provide an enhanced means of spread of the fungal pathogen due to insect transmission. This could be particularly important for fungi where mycovirus infection reduces sporulation in the fungal host [143]. With a host range that excludes dicotyledonous plants and a limited virome described to date, *F. graminearum* represents an ideal alternative host for comparative and translational research.

### 6.4. Neurospora crassa 

Honda et al. [143] have proposed *Neurospora*, as an ideal model for investigating the role of mycovirus in fungal interactions. They described mycoviruses from six RNA virus families (*Fusarivirdae*, *Partitivirdae*, *Endornaviridae*, *Reoviridae*, *Deltaflexiviridae*, and *Narnaviridae*) and an unclassified RNA virus group naturally infecting *N. crassa*, and were able to transfect spheroplasts of a *N. crassa* virus-free isolate with mycoviruses from other fungi (Rosellinia necatrix partitivirus 2 and mycoreovirus 1 from *Rosellinia necatrix* and *C. parasitica*, respectively). This organism is easy to culture and its haploid nature makes genetic manipulation relatively simple. It is already well established as a model organism with multiple cellular and genetic resources, for instance, *N. crassa* mutants were used for seminal research into quelling that described many of the proteins involved in RNAi [144,145,146,147]. *N. crassa* has been reported to infect Scots pine (*Pinus sylvestris*) and can switch between various lifestyles, from endophyte, to pathogen to saprobe depending on the host health, offering interesting biology to investigate [148]. Major limitations of *N. crassa* are its narrow plant host range and small virome described to date. Although the *B. cinerea*/mycoviral system has the advantage of infecting a broad host range of herbaceous, rather than woody hosts, *N. crassa*’s well-described RNAi machinery including genetic and cellular resources and its ability to be infected with heterologous mycoviruses make an attractive complementary fungal host. Complementary to both undertake selected experiments and an exemplar for the assembly of molecular and cellular tools to recapitulate in *B. cinerea*. 

### 6.5. Rosellinia necatrix

*Rosellinia necatrix* has a wide host range infecting over 60 genera of plants including some economically important crops and causes wilting and tree death through white root disease [149,150]. This emergent threat is present across much of the world and is regulated in some jurisdictions [151,152]. Since 2000, there are now at least 42 mycoviruses described in association with *R. necatrix*, and despite the relatively small number of groups studying mycoviruses of *R. necatrix*, major advances have been made. For instance, the description and biology of an unusual pair of mycoviruses that mimic the ‘hermit crab and the shell’ (yadokari/yadonushi); an uncoated positive-sense RNA *Yadokariviridae* virus that is encapidated within proteins made by an unrelated encapsidated *Megabirnaviridae* virus which were both isolated from *R. necatrix* [10,132,153]. Furthermore, two partitiviruses originating from *R. necatrix* have recently been shown to replicate in fungi, plants and insects, i.e., three distinct kingdoms [154]! Also, zinc chloride enhanced hyphal movement of a mycovirus between incompatible *R. necatrix* strains [155]. Although *R. necatrix* can be transformed and transfected, it is not widely used as a model fungus, perhaps because of its emergent pest nature [151]. 

### 6.6. Sclerotinia sclerotiorum 

To date *S. sclerotiorum* is host to 156 described mycoviruses, and has been at the forefront of mycovirus discovery [156,157]. The first DNA mycovirus, *Sclerotinia sclerotiorum* hypovirulence-associated DNA virus 1 (SsHADV-1), was described from *S. sclerotiorum* cultures [158] and this host-mycovirus relationship has been investigated at cellular and molecular levels [159]. The hypovirulence-associated SsHADV-1 holds promise as a biocontrol for *S. sclerotiorum* and can also enhance plant growth through a priming mechanism [1,160]. 

*S. sclerotiorum* is a highly destructive fungal pathogen with a wide host range including several staple crops and has a cosmopolitan habitat. The complete genome of *S. sclerotiorum* is sequenced and annotated [161,162]. Multiple transcriptome studies have been performed on this fungus including sRNAs sequencing that identified potential miRNAs and a subset of cleaved endogenous targets [163]. The ability to transfect *S. sclerotiorum* provides a useful method to introduce mycoviruses [157]. Disadvantages for using *S. sclerotiorum* as a model for mycovirus biology is that it is less genetically tractable than *B. cinerea. S. sclerotiorum* is a sexually reproducing ascomycete; however, it is homothallic, meaning that colonies produced from a single ascospore are self-fertile, producing apothecia with genetically uniform ascospores. As a consequence of this, genetic or mycoviral studies would be made more difficult due to the lack of outcrossing (by contrast to *B. cinerea*). *S. sclerotiorum* also lacks asexual spores (conidia), which makes pathogenicity tests both more difficult to perform, and reliably reproduce, compared with *B. cinerea*. Often mycelia (growing on agar plugs or in a liquid slurry) or ascospores are used in *S. sclerotiorum* pathogenicity tests [164,165]. Ascospores are often the main source of primary inoculum in field infections [165]. These are more difficult to produce, compared with conidia which can be grown quickly and harvested easily for use as in *B. cinerea* infection studies. 

## 7. Research Strategies to Better Understand Mycovirus Biology

Beyond the description of mycovirus diversity, a strategy is required to better understand the biology of mycoviruses, their modes of reproduction and dispersal, and the impact that they have on the physiology of their fungal host. 

It is common practice in virus research to generate constructs to transform the recipient, in this case the fungus *B. cinerea*, and express infectious viral genomes. These infectious genomes must themselves have the capacity to replicate and complete infection cycles within non-transformed isogenic fungus and recapitulate any symptom of the original mycovirus. Such infectious constructs could be modified in coding and non-coding sequences to reveal which sequence elements in the viral genome are crucial for completion of the viral replication cycle or to suppress the fungal RNAi pathways. A construct to transform infectious clones has been generated for BVF [65,166]. For multi-segment viruses, each genomic strand may be expressed from a single or distinct constructs as long as all genome components are co-transformed into the same recipient host. Co-transformation of *B. cinerea* with multiple constructs is feasible and effective, as demonstrated with up to two gene knockout constructs co-transformed and stably integrated in the genome, alongside a non-integrating autonomously replicating plasmid carrying a selection marker [30]. Furthermore, use of inducible promoters may afford control over single or multiple mycovirus replication and/or mycovirus protein expression [167].

Which host factors participate in the replication of the viral genome(s)? In order to understand the contribution of host factors to viral replication, both genetic and biochemical approaches could be exploited. By developing a viral strain containing a detectable (fluorescent?) marker, one could introduce such a marked viral strain into a recipient fungus and perform a large-scale genetic screen for fungal mutants that lose the marker expression. The defect in the host genome that results in loss of viral replication could be revealed by whole genome resequencing (perhaps using bulk segregant analysis) and identifying mutations that might be causal. Biochemical approaches include experiments to identify host proteins that interact with viral proteins, especially the viral replicase, using either co-immunoprecipitation or proximity labelling by biotinylation. Both methods are established for filamentous fungi and could be performed by expressing viral replicase proteins tagged at the C-terminus with an affinity tag (for co-IP) or a biotinylation module (Turbo-ID); [168].

What is the role of the fungal RNAi machinery in counteracting viral infection? Although it is well established that plant and animal viruses are targeted by their host RNAi machinery, little is known about this process in fungi. Is the RNAi machinery of the fungal host activated upon viral infection, and is it effective in (partially) reducing the viral replication? The expression levels of genes in the RNAi machinery could be quantified in virus-free recipients and transformants expressing an infectious viral copy. Mutants of *B. cinerea* in which both DCL genes have been knocked out, produce <1% of the level of sRNAs as compared with the wild-type [31]. Knockout mutants in the RdRp genes *Bcrdr1* and *Bcrdr2* have also been generated [99]. It would be interesting to examine the impact of such mutations on viral infections, both with respect to the viral titer and to the impact of the host physiology and development. Besides testing mutants in DCL and RDR genes, it would be relevant to generate and test mutants in AGO-like genes, of which *B. cinerea* contains four paralogs. 

If the fungal RNAi pathway is activated upon viral infection, the question arises whether the mycovirus is able to suppress the RNAi machinery. If so, it is relevant to study which viral components contribute to suppressing the RNAi machinery, and through which mechanism(s). This could be explored with viral constructs containing a quantifiable (fluorescent) marker and by generating targeted mutations in the viral genes in the infectious construct(s). It is likely that certain mycoviruses are more effective in suppressing the host RNAi machinery than others. An intriguing question will be to what extent a mycovirus that is not effective in suppressing the host RNAi machinery will benefit from co-infection with a different virus (or viruses) that is actively suppressing RNAi. Isolates of *B. cinerea* and other *Botrytis* species have been reported to contain a multitude of viruses and it is relevant to understand to what extent co-infections result in competitive or cooperative interactions between viruses. Likewise, it has been proposed that VSRs encoded by mycoviruses may contribute to the variation in outcome of applied dsRNA (exogenous or host expressed) [169]. It will be important to understand the impact of virus(es) that encode a suppressor, or in which the suppressor has been deactivated or removed, on the ability of exogenous dsRNA to target a host gene transcript for silencing. Such experiments may enable development of host and mycovirus targeted dsRNA concoctions that have improved efficacy for use in both agriculture and health sectors.

## 8. Conclusions

With a vast array of different fungi that are infected by an even larger array of different mycoviruses, we need a model fungal host in which we can deeply understand the biology of mycoviruses. Findings from the model system can then be compared and contrasted through translational research on other representative fungi that span the fungal kingdom. Knowledge from this research will enable us to understand the biology of mycoviruses and enable us to develop efficacious and durable control methods for pathogenic fungi. The highly tractable and cosmopolitan fungus *B. cinerea* provides an ideal international model for mycovirus studies. Since *B. cinerea* is a major plant pathogen, new insights may have immediate utility as well as creating new knowledge that complements and extends the knowledge of mycovirus interactions in other fungi, such as those in *S. cerevisiae, C. parasitica*, *F. graminearum*, *N. crassa*, *R. necatrix*, and *S. sclerotiorum*, alone or with their respective plant hosts. A model system for mycovirus research would also complement the study of oomycete viruses that share similar characteristics with mycoviruses.

## Figures and Tables

**Table 4 viruses-16-01483-t004:** Virus-encoded suppressor associated with mycovirus members related to those found in *Botrytis cinerea*.

Virus	VSR Protein/Mechanism	Family	Host	*Botrytis cinerea* Host?	Reference
Potato virus X	TGB1/ Blocking the silencing signal in initially infected cells or stopping its spread to uninfected cells	*Alphaflexiviridae*	Plants	No	[105]
Plantago asiatica mosaic virus					[106]
Alternanthera mosaic virus					[107]
Turnip yellow mosaic virus	p69/partially inhibits the amplification but not the execution of RNA silencing	*Tymoviridae*	Plants	No	[108]
Red clover necrotic mosaic virus	p27, p88, MP/Sequestering DCL1, potentially utilizing its helicase properties for viral replication	*Tombusviridae*	Plants	No	[109,110]
Rice grassy stunt virus	nsP5	*Phenuiviridae*	Plants	No	[111]
chikungunya virus	nsP2, nsP3	*Togaviridae*	Humans	No	[112]
Botrytis virus F	Unknown	*Gammaflexiviridae*	Fungi	Yes	[97]
Fusarium graminearum virus 1	P2 gene/FgDICER2 and FgAGO1 suppression	*Fusariviridae*	Fungi	No	[113]
Cryphonectria hypovirus 1	p29/reduction in transcription level of DCL2 and AGL2	*Hypoviridae*	Fungi	No	[114,115]
Cryphonectria hypovirus 4	p24	*Hypoviridae*	Fungi	No	[116]
Aspergillus virus 1816	unknown	*Totiviridae*	Fungi	No	[91]
Tulasnella partitivirus 2	CP	*Partitiviridae*	Fungi	No	[117]
Tulasnella partitivirus 3					

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
