# Peer review of "Mycologists and Virologists Align: Proposing Botrytis cinerea for Global Mycovirus Studies"

_viruses, 2024, doi:10.3390/v16091483_

Round 1
Reviewer 1 Report
Comments and Suggestions for Authors
The manuscript “Mycologists and virologists align: Proposing Botrytis cinerea for global mycovirus study” is a well written review/opinion paper on mycoviruses basically infecting B. cinerea, which is proposed to serve as a mycovirus research model. In this manner the manuscript differs from the recent (2023) review on mycoviruses published in Viruses by Hough et al., which could be added in the references.
Some minor points to be considered are:
- Change Genome “Polarity” in Table 1 to Genome “Type”.
- An update of the mycovirus taxonomy is needed, e.g. the current ICTV genera do not include Mitovirus or Hypovirus. They have split.
- Table 2: Change “Taxon” to “Genus”. Change “Hypovirus” to “Betahypovirus”, “Mitovirus” to “Duamitovirus” for SsMV3 and “Unuamitovirus” for SsMV4, “Botourniaviridae” to “Betascleroulivirus”.
- A reference is missing in Page 6, paragraph 4, ln. 171.
Some formatting issues follow:
- B. cinerea should be in italics throughout the manuscript (e.g. p. 2, ln. 82, 94; p. 3 ln. 97, 121)
- Table 1: Country names at the bottom should not be in italics
- P. 2, ln. 132, 133, 134, 135: Latin names should be in Italics
- P. 2, ln. 134, 135: Change “cultivar” to “cv”
- P. 7, ln. 213: Change “C. parasitica” to “Cryphonectria parasitica” (1st time mentioned)
- P. 8, ln. 265: Add “(small interfering RNAs)” after “siRNAs” (1st time mentioned)
- P. 9, ln. 300: Change “N. crassa” to “Neurospora crassa” (1st time mentioned)
- P.11, ln. 322: Delete “Alphaflexiviridae”
- P.13, ln. 417: “Fusarium” should be in italics
- P. 15, ln. 563: Change “Rdr” to “RDR”
- P. 16, ln. 588: Add “kingdom” after “fungal”
Author Response
Reviewer 1
In this manner the manuscript differs from the recent (2023) review on mycoviruses published in Viruses by Hough et al., which could be added in the references (added by authors).
Some minor points to be considered are:
- Change Genome “Polarity” in Table 1 to Genome “Type”. (addressed by authors)
- An update of the mycovirus taxonomy is needed, e.g. the current ICTV genera do not include Mitovirus or Hypovirus. (addressed in both main text and Table S1 by authors).
- Table 2: Change “Taxon” to “Genus”. (addressed by authors)
- Change “Hypovirus” to “Betahypovirus”, “Mitovirus” to “Duamitovirus” for SsMV3 and “Unuamitovirus” for SsMV4, “Botourniaviridae” to “Betascleroulivirus”. (addressed in both main text and Table S1 by authors).
- A reference is missing in Page 6, paragraph 4, ln. 171. (addressed by authors).
Some formatting issues follow:
- B. cinerea should be in italics throughout the manuscript (e.g. p. 2, ln. 82, 94; p. 3 ln. 97, 121) (addressed by authors)
- Table 1: Country names at the bottom should not be in italics (addressed by authors)
- P. 2, ln. 132, 133, 134, 135: Latin names should be in Italics (addressed by authors)
- P. 2, ln. 134, 135: Change “cultivar” to “cv” (addressed by authors)
- P. 7, ln. 213: Change “C. parasitica” to “Cryphonectria parasitica” (1st time mentioned) (addressed by authors)
- P. 8, ln. 265: Add “(small interfering RNAs)” after “siRNAs” (1st time mentioned) (addressed by authors)
- P. 9, ln. 300: Change “N. crassa” to “Neurospora crassa” (1st time mentioned) (addressed by authors)
- P.11, ln. 322: Delete “Alphaflexiviridae” (addressed by authors)
- P.13, ln. 417: “Fusarium” should be in italics (addressed by authors)
- P. 15, ln. 563: Change “Rdr” to “RDR” (addressed by authors)
- P. 16, ln. 588: Add “kingdom” after “fungal” (addressed by authors)
Reviewer 2 Report
Comments and Suggestions for Authors
This is an interesting and timely review on mycoviruses, which proposes that more focus should be placed on using gray mold (Botrytis cinerea), an important phytopathogen, and its various mycoviruses as models. The review is very up to date and proposes new avenues of work to fill important knowledge gaps in the field (Section 7). It will be a useful addition to the literature, and is clearly written which I appreciated.
I have no major criticisms, only minor, as follows.
- The Latin name of the fungus is inconsistently italicized in the first few paragraphs
- Line 459, remove comma after ‘crassa’
- Table S1, the excel sheet with the current listing of Botrytis viruses, should be referred to somewhere in the text, and its importance/relevance to the article explained.
Author Response
Reviewer 2
I have no major criticisms, only minor, as follows.
- The Latin name of the fungus is inconsistently italicized in the first few paragraphs (addressed by authors)
- Line 459, remove comma after ‘crassa’ (addressed by authors)
- Table S1, the excel sheet with the current listing of Botrytis viruses, should be referred to somewhere in the text, and its importance/relevance to the article explained. (addressed by authors)
We have included reference to a new publication on lines 206-208. All references have updated numbering as required to accommodate the additional references.